# Highly Porous-Cellulose-Acetate-Nanofiber Filters Fabricated by Nonsolvent-Induced Phase Separation during Electrospinning for PM_2.5_ Capture

**DOI:** 10.3390/nano12030404

**Published:** 2022-01-26

**Authors:** Sang-Hyun Ji, Ji-Sun Yun

**Affiliations:** Energy & Environmental Division, Korea Institute of Ceramic Engineering and Technology, Jinju 52851, Korea; sanghyun_ji@kicet.re.kr

**Keywords:** nonsolvent-induced phase separation, porous nanofiber, cellulose acetate, PM_2.5_ capture, electrospinning

## Abstract

Highly porous-cellulose-acetate (CA) nanofibers were prepared by an electrospinning process based on a nonsolvent-induced phase separation (NIPS) mechanism, and their PM_2.5_ capture efficiencies were evaluated. The NIPS condition during the electrospinning process was achieved by selecting appropriate good and poor solvents based on the Hansen solubility parameters of CA. *N*,*N*-dimethylacetamide (DMAc) was used as the good solvent, while dichloromethane (DCM), tetrahydrofuran (THF), and acetone were used as poor solvents. Porous-CA nanofibers were observed upon using the binary solvent systems of DCM:DMAc = 1:9, DCM:DMAc = 2:8, and THF:DMAc = 1:9, and the CA nanofibers formed using the DCM/DMAc system with DCM:DMAc = 1:9 were found to have the highest specific surface area of 1839 m^2^/g. Based on the optimized binary solvent system with DCM:DMAc = 1:9, porous-CA nanofibers were prepared and characterized according to the CA content in the electrospinning mixture. The results confirmed that a porous structure was formed well from the surface to the core of the nanofibers. The composition range of the ternary mixture of CA and two solvents capable of producing porous-CA nanofibers was mapped on a ternary phase diagram, and highly efficient PM_2.5_ capture with 98.2% efficiency was realized using porous-CA nanofibers obtained using a 10 wt.% CA solution. This work provides a new strategy for improving the efficiency of porous-nanofiber filters for PM_2.5_ capture.

## 1. Introduction

Among the recently emerged air pollutants, fine particulate matter (PM) has serious adverse effects on the quality of life and health of human beings [1,2]. According to the Global Burden of Disease project report, in 2016, long-term exposure to PM with aerodynamic diameters less than 2.5 μm (referred to as PM_2.5_) decreased the average life expectancy at birth by approximately 1 year and, particularly, by approximately 1.2–1.9 years in polluted countries, such as those in Asia and Africa [3]. Although PM pollutants usually originate from outdoor emissions, such as from industrial sites and automobiles, and many combustion processes, outdoor PM pollutants considerably affect the indoor air quality [4,5]. Furthermore, because most individuals spend a large part of their time indoors, it is important to reduce and control indoor PM pollutants, especially PM_2.5_ [6]. Although two types of air filters, viz., porous-membrane filters and thick-fibrous air filters, are commonly used for the removal of PM_2.5_, these filters cause serious issues, such as a significant air-pressure drop and limited PM_2.5_-removal efficiency [7,8]. In this regard, there is a need to develop a new type of filter that can capture PM_2.5_ with high efficiency.

To find the most effective material and structure for highly efficient PM_2.5_ capture, various electrospun polymer nanofibers based on various polymers, such as polyacrylonitrile (PAN), polyvinylpyrrolidone, polystyrene, polyvinyl alcohol, polypropylene, and polyimide (PI), have been studied [7,8]. According to a previous study, PM pollutants first wrap around the PAN nanofibers, and the PM particles move along the PAN nanofibers and aggregate to form larger particles, leaving some empty spaces for the capture of new incoming PM particles [7]. In order to remove high-temperature PM pollutants, a study with PI nanofibers with high-heat resistance was also conducted [8]. However, to enable ecofriendly treatment of filters adsorbed with PM pollutants as wastes, it is necessary to develop highly efficient nanofibers derived from natural materials, such as wood and cotton. Cellulose acetate (CA), a bioplastic, is usually extracted from wood pulp or cotton yarn and is used in a wide range of applications, such as in tissue engineering, wound dressing, and separation membrane [9]. As filters based on CA are biodegradable after use, ecofriendly waste treatment is feasible. Furthermore, CA nanofibers manufactured by electrospinning have attractive advantages, such as high-filtering efficiency, low air resistance, light weight, and biocompatibility [7,10,11,12]. However, to maximize the ability of CA nanofiber filters to capture PM_2.5_, it is necessary to improve their specific surface area through structural design.

In this study, highly porous electrospun CA nanofibers were fabricated by an electrospinning process based on a nonsolvent-induced phase separation (NIPS) mechanism using mixed solvent systems based on solvents with different CA solubilities and boiling points. The phase separation mechanism is initiated as the binary solvent system undergoes a transition from a stable single-phase solution to an unstable biphasic mixture owing to the difference in the solvent evaporation rates of the two solvents [13]. The volatile solvent evaporates mainly from the surface of the jet during the electrospinning process, and the solvent molecules diffuse from the core to the surface. Meanwhile, polymer molecules diffuse from the jet surface to the core, but the polymer diffusion is usually hindered by the nonvolatile solvent remaining at an increased concentration. The formation of heterogeneous solvent domains within the electrospinning jet leads to irregular pores in the nanofibers. In this study, *N*,*N*-dimethylacetamide (DMAc) was chosen as a good solvent for CA, while dichloromethane (DCM), tetrahydrofuran (THF), and acetone were chosen as poor solvents. The binary solvent mixtures consisting of the good and one of the poor solvents were formulated at different solvent ratios, and porous-CA nanofibers were prepared by varying the CA content in the electrospinning mixture. The formation of a porous structure from the surface to the core of the nanofibers was observed, and a ternary phase diagram was constructed to determine the CA/good solvent/poor solvent composition range over which porous-CA nanofibers are produced. Furthermore, possible correlation between the specific surface area of the porous-CA nanofibers and the efficiency of PM_2.5_ capture was also investigated according to the CA content. This study provides new possibilities for improving the efficiency of biodegradable porous-nanofiber filters for PM_2.5_ capture.

## 2. Material and Methods

CA (average molecular weight = ~30,000, Sigma-Aldrich, St. Louis, MO, USA) was heated at the rate of 10 °C/min to 350 °C and calcined at 350 °C for 1 h and then sieved with a mesh to adjust the particle size of the powder to 100 μm or less. To prepare binary solvent mixtures, a poor solvent (DCM (Sigma-Aldrich, St. Louis, MO, USA), THF (Sigma-Aldrich, St. Louis, MO, USA), or acetone (purity = 99.995%, Daejung Chemical, Siheung, Korea)) and a good solvent (DMAc (Sigma-Aldrich, St. Louis, MO, USA)) were vigorously mixed at different volume ratios of 9:1, 8:2, 7:3, 6:4, and 5:5 for 24 h at 500 rpm. Different CA precursor solutions for the electrospinning process were prepared by adding 8, 10, or 12 wt.% of the calcined CA powder to the binary solvent mixtures, followed by stirring at 500 rpm for 24 h.

The CA precursor solution was loaded into a 10 mL plastic syringe fitted with a 20 G metallic needle, and the needle was connected to a high-voltage supply unit. The voltage supplied for electrospinning was 10–15 kV, and the feed rate of the spinning solution was 1.0 mL/h. The distance between the tip of the syringe and rotary collector was 10 cm, and the collector was rotated at a speed of 100 rpm. All the electrospinning processes were carried out at room temperature, and the humidity was maintained at 40%. The CA nanofibers were finally dried at 80 °C for 24 h to remove any residual solvent.

The CA nanofibers were characterized using a field-emission scanning electron microscope (FE-SEM, JSM-6700F, JEOL, Akishima, Japan) and a micropore physisorption analyzer (ASAP 2020M, Micromeritics, Norcross, GA, USA). The specific surface area was calculated using the Brunauer–Emmett–Teller (BET) equation. The volumes of the micropores and mesopores were calculated using the t-plot method and Barrett–Joyner–Halenda (BJH) method, respectively. The removal efficiency of PM (E) was evaluated using an aerosol generator (QRJZFSQ-I, Beijing, China) and a particle counter (EPAM-5000, SKC Inc., Covington, LA, USA) in FITI Testing and Research Institute. PMs with various sizes of 0.5 μm, 1.0 μm, and 2.5 μm were supplied to the filter with air permeability of 290–310 cm^3^/cm^2^/s at a rate of 10 L/min for 8 h, and the experimental environment was maintained at a temperature of 25 °C and a humidity of 25%. The PM-removal efficiency was calculated using the following equation:(1)$$E\;(\%)=\left(\frac{C_{u}-C_{d}}{C_{u}}\right)\times 100$$
where C_u_ and C_d_ are the concentrations of PM before and after adsorption using the air nanofilters, respectively.

## 3. Results and Discussion

In the fabrication of CA nanofibers using the NIPS mechanism, solvent properties, such as the boiling temperature and solubility, and whether the chosen solvent is a good one for CA are important factors. The NIPS condition during the electrospinning process is initiated by the difference in the evaporation rates of the good solvent and poor solvent in the binary solvent mixture [10,14]. In order to maximize the difference between the evaporation rates of the good and bad solvents, materials with a boiling temperature of 100 °C or higher were considered good solvents, and those with a boiling temperature of 70 °C or lower were considered bad solvents. Furthermore, Hansen solubility parameters (HSPs) are suitable for assessing suitable dissolution conditions of CA in various solvents. HSPs can be used to estimate the affinity between the polymer and a given solvent and between two solvents in a binary mixture [15]. The difference (R_a_) between the HSPs of two materials can be calculated using the following equation [16,17]:(2)$$R_{a}=\sqrt{4{\left(\delta_{d1}-\delta_{d2}\right)}^{2}+{\left(\delta_{p1}-\delta_{p2}\right)}^{2}+{\left(\delta_{h1}-\delta_{h2}\right)}^{2}}$$
where δ_d_, δ_p_, and δ_h_ represent the dispersion, polar, and hydrogen bond interactions, respectively. The smaller the R_a_, the higher the similarity of the HSPs of two materials, indicating better affinity between them. Considering these characteristics, in this study, DMAc was chosen as a good solvent, while DCM, THF, and acetone were chosen as bad solvents (Table 1). Specifically, by calculating the R_a_ values of various solvents in relation to CA, DMAc with the lowest R_a_ value of 5.42 and a boiling temperature of 165.5 °C was determined to be a good solvent. On the other hand, DCM, THF, and acetone, which yield higher R_a_ values of 8.69, 6.00, and 6.63, respectively, indicating that their affinity for CA is lower than that of DMAc, were determined to be bad solvents. Furthermore, because the boiling temperatures of DCM (39.8 °C), THF (66 °C), and acetone (56.3 °C) are lower than that of DMAc (165.5 °C), they are expected to be advantageous for phase separation.

The microstructures and diameter distributions of porous-CA nanofibers fabricated using 10 wt.% CA solutions were studied according to different compositions of various binary solvent mixtures consisting of the good and poor solvents (Figure 1). When the binary solvent system of DCM/DMAc, with solvents having different R_a_ values and boiling temperatures, was used for electrospinning, thick- and porous-CA nanofibers with an average diameter of ~3 μm were observed at DCM:DMAc ratios of 1:9 and 2:8. In general, as the amount of the good solvent, DMAc, increased (or as the amount of the bad solvent, DCM, decreased), more pores were formed, and the fiber diameter increased and the diameter distribution widened [25]. Due to the differences between the boiling temperatures and solubilities of DCM and DMAc are large, DCM on the surface of the electrospinning jet possibly volatilized more quickly, and phase separation was accomplished adequately. The formation of heterogeneous regions within the jet of the electrospinning solution resulted in irregular pores in the CA nanofibers [10,14], making it possible to generate porous-CA nanofibers. However, as the amount of DCM increased, the distribution of the heterogeneous regions became uniform from the surface to core of the jet, resulting in the formation of thin compact nanofibers of a uniform diameter instead of porous nanofibers. When the binary solvent systems of THF/DMAc and acetone/DMAc were used, only a few pores were observed on the surface only in the CA nanofibers obtained using THF/DMAc at THF: DMAc = 1:9. However, it was difficult to determine if pores had developed inside the fibers due to the thinness of the fibers with an average diameter of 1.8 μm. Furthermore, in the CA nanofibers obtained using acetone/DMAc at acetone:DMAc = 3:7 and 4:6, electrospinning could not be performed properly, and only beads were mainly observed. The binary solvent systems of THF/DMAc and acetone/DMAc are not suitable for CA-fiber fabrication because the differences in the boiling points and HSPs between the solvents are not large enough to induce the NIPS condition.

Figure 2a shows the N_2_ adsorption/desorption isotherms of the CA nanofibers according to the various binary solvent mixtures at a poor solvent:DMAc volume ratio of 1:9. The specific surface areas of the CA nanofibers fabricated using the different binary solvents of DCM/DMAc, THF/DMAc, and acetone/DMAc were determined to be 1839, 743, and 514 m^2^/g, respectively. The DCM/DMAc-based CA nanofibers, in which pores were most clearly formed (Figure 1), had the highest specific surface area. The specific surface area of the THF/DMAc-based CA nanofibers with some pores was significantly lower, while the acetone/DMAc-based CA nanofibers with no pores had the lowest specific surface area. The results indicate that the binary solvent system of DCM/DMAc was suitable for NIPS, and, consequently, it was possible to prepare porous-CA nanofibers with a high specific surface area. Further, the N_2_ adsorption/desorption isotherms of the DCM/DMAc-based CA nanofibers were studied according to different DCM:DMAc ratios (Figure 2b). The specific surface areas of the DCM/DMAc-based CA nanofibers obtained at DCM:DMAc volume ratios of 1:9, 2:8, 3:7, and 4:6 were 1839, 1435, 743, and 514 m^2^/g, respectively. As the amount of DCM increased, the specific surface area of the DCM/DMAc-based CA nanofibers decreased significantly because the porosity of the fiber (Figure 1) decreased. Based on this result, the optimum composition of the DCM/DMAc binary solvent system for preparing porous-CA nanofibers based on NIPS was determined to be DCM:DMAc = 1:9.

Based on the optimized binary solvent system with DCM:DMAc = 1:9, the microstructures of the DCM/DMAc-based CA nanofibers were investigated according to the CA content (Figure 3). The microstructures of CA nanofibers obtained using solutions with 8 and 10 wt.% CA revealed that a porous structure was well developed from the surface to the core of the nanofibers. However, the CA nanofibers obtained using a 12 wt.% CA solution showed compact nanofibers without any pores. During the NIPS process, in addition to solvent molecules, polymer molecules also diffuse from the jet surface to the core owing to phase separation induced by the nonuniform evaporation of solvents. However, the polymer diffusion is hindered by the residual nonvolatile solvent and polymer units [9]. The porous-fiber morphology is caused by the formation and subsequent collapse of a solid polymer skin on the surface of the jet. However, when the polymer content is high, pores are not likely to form because the collapse of the polymer skin is difficult. Therefore, porous nanofibers were only formed over a certain range of CA concentration, and pore formation was difficult at a high CA concentration of 12 wt.%.

Further, the N_2_ adsorption/desorption isotherms of the DCM/DMAc-based CA nanofibers formed at different CA contents were studied (Figure 4a). Porous-CA nanofibers obtained using 8 and 10 wt.% CA solutions (Figure 3) had higher specific surface areas of 1747 and 1839 m^2^/g, respectively. On the other hand, nonporous-CA nanofibers formed using a 12 wt.% CA solution had a lower specific surface area of 798 m^2^/g. The BJH pore-size distribution of the CA nanofibers is shown in Figure 4b. In the case of the CA nanofibers with the highest specific surface area obtained using a 10 wt.% CA solution, the largest number of micropores was developed with pore sizes in the range of 0.5–1.0 nm. On the other hand, in the case of CA nanofibers with the lowest specific surface area obtained using a 12 wt.% CA solution, the micropores were developed only to a small extent. We also investigated the correlation between the efficiency of PM_2.5_ capture and the specific surface area of the porous-CA nanofibers obtained using solutions with different CA contents (Figure 4c). The CA nanofibers with the highest specific surface area and the most well-developed micropores obtained using a 10 wt.% CA solution exhibited the highest PM_2.5_ capture efficiency of 98.2%, while the PM_2.5_ capture efficiency of the CA nanofibers obtained using an 8 wt.% CA solution was slightly lower at 93.1%. On the other hand, the efficiency of PM_2.5_ capture for the CA nanofibers with the lowest specific surface area obtained using a 12 wt.% CA solution was the lowest at 66.6%. These results clearly suggest that the better the pore development, the higher the specific surface area of the fiber and the higher the efficiency of PM_2.5_ capture. In the case of the CA nanofibers with the highest efficiency of PM_2.5_ capture obtained using a 10 wt.% CA solution, as the size of the PM pollutants decreased from PM_2.5_ to PM_1.0_ and then to PM_0.5_, the efficiency of PM capture decreased from 98.2% to 93.2% and then to 73.6%, respectively. In other words, the efficiency of PM capture by the CA nanofiber filter decreased as the size of the PM pollutant decreased, owing to the size effect. However, even very small particles of PM_0.5_ could attach to the porous-CA nanofiber surface, and they were able to move along the CA nanofibers and aggregate to form larger particles [7], even when the pore size of the filter was larger than the size of the PM pollutant, resulting in a high efficiency of PM_0.5_ capture of 73.6%. The composition range over which the porous-CA nanofibers were produced was mapped on a ternary phase diagram, as shown in Figure 4d. For the ternary system of CA, DCM (poor solvent), and DMAc (good solvent), it was confirmed that porous-CA nanofibers could be formed well in the DCM:DMAc range of 1:9 to 2:8 at 8–10 wt.% CA content. The diagram indicates the composition range over which the ternary system can produce highly porous bead-free CA nanofibers.

## 4. Conclusions

Highly porous-electrospun-CA-nanofiber filters were fabricated based on the NIPS mechanism. For this, mixtures of a good solvent (DMAc) and poor solvents (DCM, THF, and acetone) with different CA solubilities and boiling points were evaluated as binary solvents. In the case of the binary solvent system of DCM/DMAc, highly porous-CA nanofibers were observed at DCM:DMAc = 1:9 and 2:8, and in the cases of THF/DMAc and acetone/DMAc systems, porous-CA nanofibers were only observed with the THF/DMAc system at THF:DMAc = 1:9. The porous DCM/DMAc-based CA nanofibers obtained at DCM:DMAc = 1:9 had the highest specific surface area of 1839 m^2^/g, indicating the optimum solvent composition to be DCM:DMAc = 1:9. Based on the optimized binary solvent system, porous-CA nanofibers were prepared by varying the CA content in the electrospinning mixture, and the highest specific surface was observed at a CA content of 10 wt.%. Furthermore, the porous structure of the CA nanofibers obtained using a 10 wt.% CA solution was well developed from the surface to the core, and highly efficient PM_2.5_ capture (efficiency of 98.2%) was observed. A ternary phase diagram that indicates the composition range over which the ternary system of CA, DCM (poor solvent), and DMAc (good solvent) can produce highly porous-CA nanofibers was also constructed. This study provides new insights into the formation of porous-nanofiber filters for highly efficient PM_2.5_ capture.

## Figures and Tables

**Figure 1 nanomaterials-12-00404-f001:**
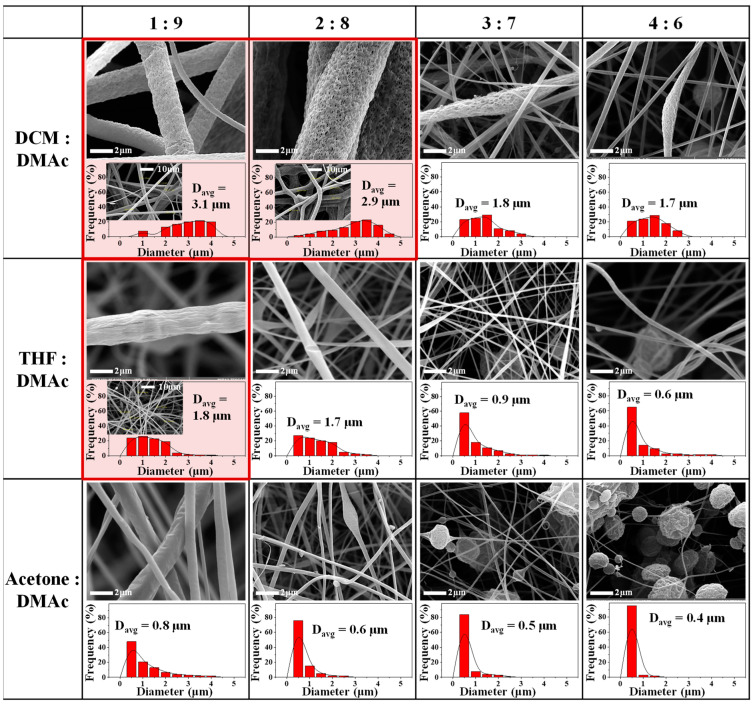
Scanning electron micrographs and histograms showing the diameter distribution of CA nanofibers according to the compositions of different binary solvent mixtures consisting of a good solvent and poor one.

**Figure 2 nanomaterials-12-00404-f002:**
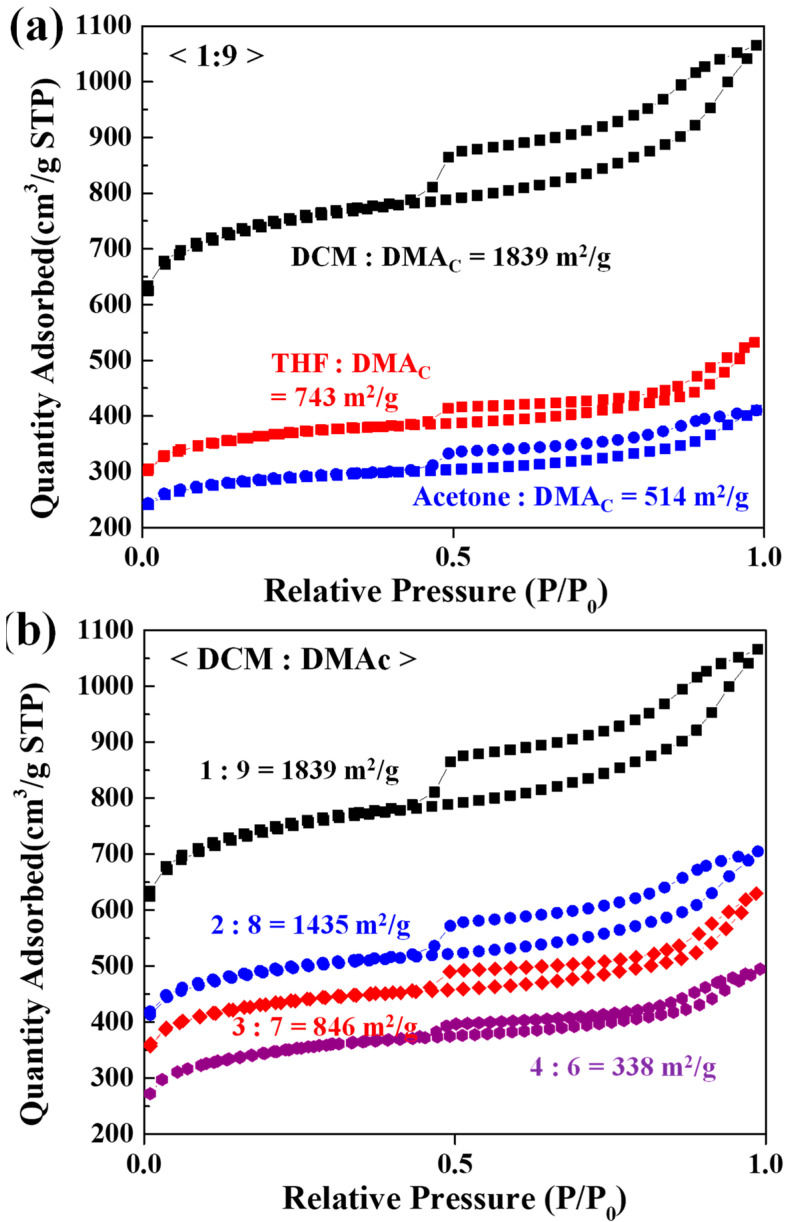
N_2_ adsorption/desorption isotherms of electrospun CA nanofibers according to (**a**) different poor solvents in binary solvent mixtures (DCM/DMAc, THF/DMAc, and acetone/DMAc) at a poor solvent:DMAc volume ratio of 1:9 and (**b**) different DCM:DMAc ratios.

**Figure 3 nanomaterials-12-00404-f003:**
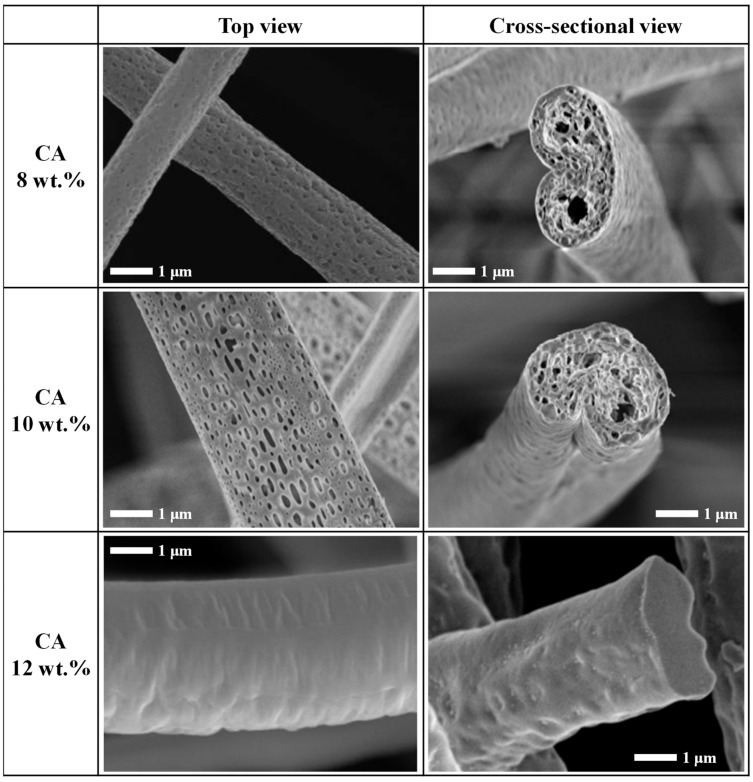
Scanning electron micrographs of DCM/DMAc-based CA nanofibers at different CA contents in the electrospinning mixture.

**Figure 4 nanomaterials-12-00404-f004:**
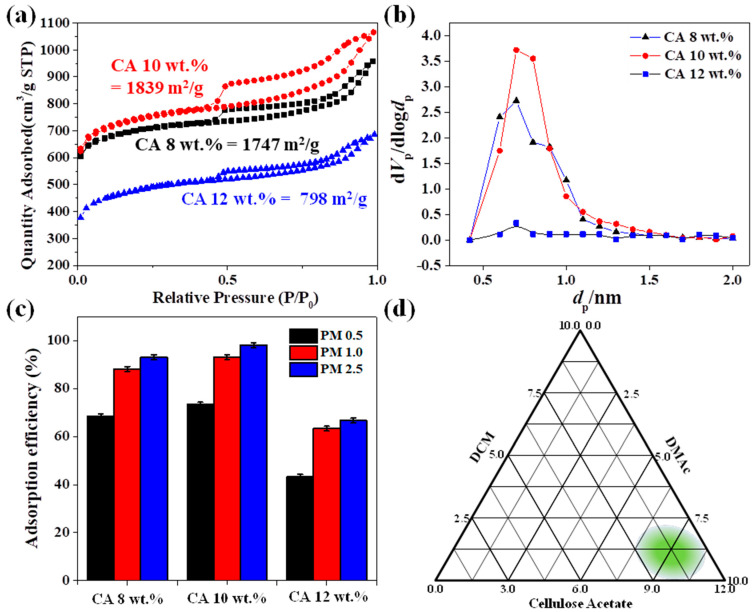
Characteristics of the DCM/DMAc-based CA nanofibers according the CA content: (**a**) N_2_ adsorption/desorption isotherms, (**b**) BJH pore-size distribution, (**c**) efficiency of PM_2.5_ capture, and (**d**) ternary phase diagram plotted to determine the composition range for producing porous-CA nanofibers.

**Table 1 nanomaterials-12-00404-t001:** Properties of the solvents and CA used in this work.

	Boiling Temperature (°C)	Dielectric Constant	Surface Tension at 20 °C (mN/m)	Hansen Solubility Parameters				Refs.
				δ_d_	δ_p_	δ_h_	R_a_	
DCM	39.8	8.93	28.1	18.2	6.3	6.1	8.69	[18,19]
THF	66	7.47	26.4	16.8	5.7	8.0	6.00	[19,20,21]
Acetone	56.3	20.7	25.2	15.5	10.4	7.0	6.63	[20,22]
DMAc	165.5	37.8	36.7	16.8	11.5	10.2	5.42	[20,21,23]
CA	-	-	-	16.0	7.5	13.5	-	[24]

## Data Availability

Data presented in this study are available by requesting from the corresponding author.

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
