# Peer review of "Highly Porous-Cellulose-Acetate-Nanofiber Filters Fabricated by Nonsolvent-Induced Phase Separation during Electrospinning for PM2.5 Capture"

_nanomaterials, 2022, doi:10.3390/nano12030404_

Round 1

Reviewer 1 Report

In this manuscript entitled “Highly porous cellulose acetate nanofiber filters fabricated by nonsolvent-induced phase separation during electrospinning for PM2.5 capture”, a porous cellulose acetate nanofiber membrane was successfully prepared by electrospinning to capture PM2.5 and purify air. However, there are still some issues to be addressed. It needs revisions and improvements before the acceptance.

  1. In the second paragraph of Introduction, although the author lists the fiber membranes formed by various polymer electrospinning, there are only two references cited. Too few references cannot explain the research progress in this field in recent years. I recommend that the author refer to the following publication or others to improve the content of this part. DOI: 10.1016/j.jobab.2020.10.001;
  2. The authors should motivate the readers why they choose electrospinning technology to fabricate aligned micro/nanofibrous membranes and show the clear advances of this technology and the following article or others should be interesting: DOI: 10.1038/s41565-021-00976-3.
  3. In the second section ' Materials and Methods ', the author lacks many classical characterizations to prove the properties of the electrospun membranes, such as XRD, TGA, FTIR and so on. It should be noted that when referring to instruments and equipment such as scanning electron microscope in this section, brackets should include their abbreviations, and in the following text, they should be replaced by abbreviations to better read and understand. You can refer to the following publication to improve its characterization: DOI: 10.1016/j.jobab.2020.03.003.
  4. In addition to the filtration efficiency, the quality factor Qf is also an important calculation parameter that the author should not ignore. Otherwise, how should the author evaluate the overall performance of the electrospinning filter in terms of removal efficiency and pressure drop ?
  5. All the histograms should be added with error bars as much as possible.

Reviewer 2 Report

Porous microfibers and what appear to be solid nano fibers were both produced in the NIPS method described.  It is not clear whether the filtration effect is mainly derived from the nano fibers or the porous microfibers.   The relationship between fiber size distribution and filtration is not fully established to demonstrate the effectiveness (if any) of the porous fibers towards the capture of particulates.  

Reviewer 3 Report

This work reports the synthesis of highly porous cellulose acetate (CA) nanofibers by an electrospinning process based on a nonsolvent-induced phase separation (NIPS) mechanism for PM2.5 capture. mixtures of a good solvent (DMAc) and poor solvents (DCM, THF, and acetone) with different CA solubilities and boiling points were evaluated as binary solvents. The porous DCM/DMAc-based CA nanofibers obtained at DCM : DMAc = 1 : 9 had the highest specific surface area, which was demonstrated for highly efficient PM 2.5 capture.

This paper is well organized. I would like to recommend the publication of this manuscript after addressing the following concerns:

  • The author spends a lot of time on how to obtain nanofibers with high specific surface area, but the discussion on filtration is very poor. More experimental details on filtration should be added.
  • In evaluating the filtration efficiency of air filters, besides filtration efficiency, pressure drop and quality factor are widely accepted as the most important parameter, which should be added and discussed.
  • As an air filter material, mechanical strength is an important factor that must be considered. The mechanical strength of the filter material obtained at different experiment condition should be added.
  • The analysis of PM filtration results (such as SEM, FT-IR, XPS of the filter sample before and after filtration) and the filtration mechanism should be further discussed.

Reviewer 4 Report

The current work proposed a method that can fabricate highly porous cellulose acetate nanofibers and demonstrated their good performance of being used as PM filters. Characteristic data regarding the effects of the different solvent systems on the morphologies of fibrous mats were presented. Overall, I recommend that this paper be accepted by Nanomaterials after major revision.

  1. The necessity of developing the proposed porous filters, as well as the uniqueness and innovativeness, which were lacking in the current version of the manuscript, should be better stated in the introduction part.
  2. Figure 1 showed SEM images and diameter distribution histograms of CA nanofibers fabricated via various solvent mixtures. However, it would be better to replace the SEM images with the ones that match the corresponding frequency distributions, not just show a single fiber in the image. Meanwhile, adjusting the position of scale bars in all images to make them more uniform is necessary.

Round 2

Reviewer 1 Report

The authors answered well the raised questions. The current version could be accepted for the publication. 

Author Response

Thank you for your kind comments. It is an honor for me to have the opportunity of our manuscript published in Nanomatierials.

Reviewer 4 Report

The authors have revised the manuscript according to reviewers' comments. I would like to recommend the acceptance of this manuscript.

Author Response

(The authors gave the same response as above.)
